# Bring Reason to Vision:
# Understanding Perception and Reasoning through Model Merging

**Shiqi Chen** [* 1]  **Jinghan Zhang** [* 2]  **Tongyao Zhu** [3]  **Wei Liu** [2]  **Siyang Gao** [1]  **Miao Xiong** [3]  **Manling Li** [4]
**Junxian He** [2]

## Abstract

Vision-Language Models (VLMs) combine visual perception with the general capabilities, such as reasoning, of Large Language Models (LLMs). However, the mechanisms by which these two abilities can be combined and contribute remain poorly understood. In this work, we explore to compose perception and reasoning through model merging that connects parameters of different models. Unlike previous works that often focus on merging models of the same kind, we propose merging models *across modalities*, enabling the incorporation of the reasoning capabilities of LLMs into VLMs. Through extensive experiments, we demonstrate that model merging offers a successful pathway to transfer reasoning abilities from LLMs to VLMs in a *training-free* manner. Moreover, we utilize the merged models to understand the internal mechanism of perception and reasoning and how merging affects it. We find that perception capabilities are predominantly encoded in the early layers of the model, whereas reasoning is largely facilitated by the middle-to-late layers. After merging, we observe that all layers begin to contribute to reasoning, whereas the distribution of perception abilities across layers remains largely unchanged. These observations shed light on the potential of model merging as a tool for multimodal integration and interpretation. Our code is publicly available at: https://github.com/shiqichen17/VLM_Merging.

## 1. Introduction

Multimodal reasoning is crucial for a variety of important applications, such as interpreting charts and figures in scientific publications and government reports. Despite the great successes of Vision-Language Models (VLMs) in tasks requiring perceptual and linguistic integration (Li et al., 2024; Liu et al., 2024; Bai et al., 2023; Wang et al., 2024b), these models struggle with complex multimodal reasoning tasks (Lu et al., 2024; Zhang et al., 2024b). This limitation – partly due to the scarcity of multimodal reasoning data – leaves them lagging far behind their language model counterpart which has made remarkable advancement in reasoning tasks (Yang et al., 2024; DeepSeek-AI et al., 2025).

Perception and reasoning are two fundamental components in this context. While language models primarily represent the reasoning ability, VLMs demand both to succeed. Therefore, it is natural to ask: can we incorporate the reasoning ability of LMs into VLMs? Achieving such a combination is challenging, as the interaction between perception and reasoning within VLMs remains poorly understood. In this work, we investigate these questions through the lens of model merging (Ilharco et al., 2023), a straightforward approach to explore whether perception and reasoning can be combined across modalities and how these two abilities are embedded within VLMs.

Concretely, model merging generates a new model by performing arithmetic operations on the parameters of existing models, without requiring additional training. This strategy works based on the assumption that models fine-tuned from a shared initialization reside in a connected subspace of the parameter space. While previous works on model merging focus on models of the same kind (Yadav et al., 2023; Yu et al., 2024b), it remains unknown that whether models across different modalities are connectable to yield benefits.

In this work, we specifically focus on the textual components of the VLMs and select LLMs with task-specific reasoning abilities that match the VLM's configuration, performing a weighted average operation on their parameters as demonstrated in Figure 1.

We conduct extensive experiments by merging commonly

---

[*]Equal contribution  [1]City University of Hong Kong [2]Hong Kong University of Science and Technology [3]National University of Singapore [4]Northwestern University. Correspondence to: Shiqi Chen <schen438-c@my.cityu.edu.hk>, Jinghan Zhang <jzhangjv@cse.ust.hk>, Junxian He <junxianh@cse.ust.hk>.

*Proceedings of the $42^{nd}$ International Conference on Machine Learning*, Vancouver, Canada. PMLR 267, 2025. Copyright 2025 by the author(s).

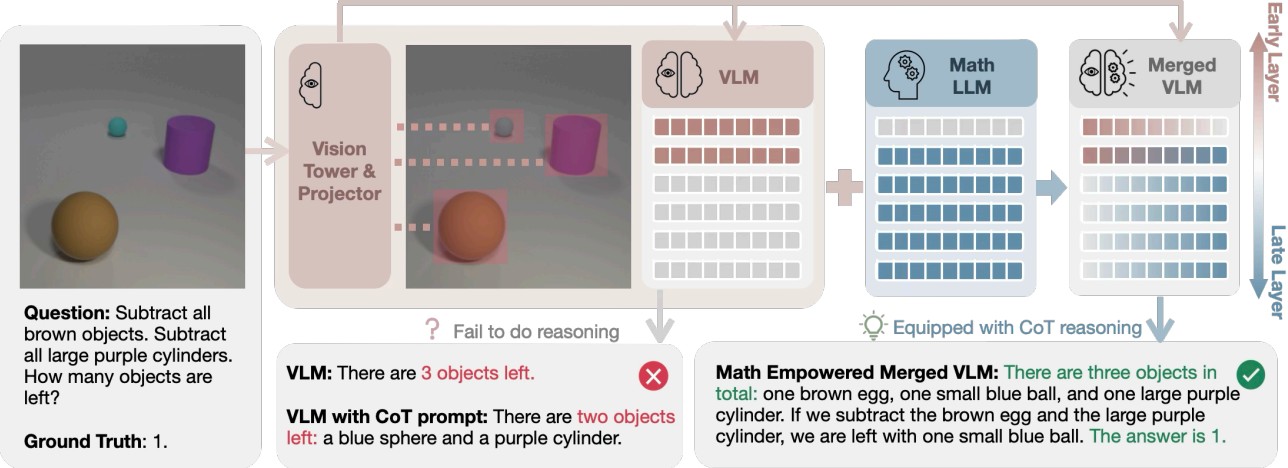

Figure 1: Illustration of our work investigating how model merging works when transferring reasoning ability from a Math-specific LLM to the VLM, showcasing the effects and results of the merged model, as well as the interpretation of layer-wise abilities. The abilities are represented in corresponding colors: **red** indicates **perception ability hidden in early layers**, while **blue** denotes **reasoning ability hidden in relatively later layers**.

used VLMs with LLMs trained on diverse math reasoning datasets (§4). Our findings demonstrate that model merging consistently improves the reasoning capabilities of VLMs across all math benchmarks, with minimal impact on the perception-dominant tasks. For instance, merging with Dart (Tong et al., 2024) enhances LLaVA's performance on MathVista's (Lu et al., 2024) math-related subset, yielding a 3.6-point absolute improvement. Even in **Vision-Only mode** of MathVerse (Zhang et al., 2025), where questions are presented in images, it also achieves a 1.4-point absolute improvement. Experiments on multiple reasoning-related benchmarks and various VLMs consistently show improvement. These results underscore the potential of parameter merging as a simple yet powerful mechanism for capability transfer across model architectures.

Furthermore, we delve deeper into understanding how model merging works and how perception and reasoning interplay in this context. Specifically, we analyze parameter changes during the merging process to investigate whether different capabilities, such as high-level reasoning and low-level perception, are disentangled within distinct subspaces of the model's parameter space (§5). Through knockout analysis, we identify two key observations: (1) Image perceptual abilities and world knowledge are predominantly embedded in the early layers of the model, whereas mathematical reasoning skills are concentrated in the middle-to-late layers; and (2) Merging with reasoning models brings reasoning abilities to all the layers, while having a minimal impact on the layer distribution of perception ability. These findings contribute to a better understanding of how reasoning can be transferred between models and provide insights into model compositionality, offering a promising approach

for enhancing multi-modal reasoning systems.

## 2. Model Merging Across Modalities

In this section, we introduce model merging for transferring the reasoning abilities of textual LMs to VLMs. A typical VLM consists of three key components: a vision tower, a language Model, and a projector that bridges these two parts. The vision tower processes images, enabling the model to "see" visual content, while the language model serves as the reasoning engine, processing knowledge and generating responses. Therefore, we target the language model ($\theta_{\text{vlm}}$) for merging while keeping the vision tower and projector unchanged.

Model merging has emerged as a promising "free lunch" technique, enabling performance improvements by reusing existing models through simple arithmetic operations on their parameters, without requiring additional training. For simplicity, we adopt *linear merging* (Ilharco et al., 2023), a widely used and robust merging strategy, in our main experiments. We also experiment with TIES merging (Yadav et al., 2023) in some cases to compare both methods in Appendix C.

The core idea of model merging relies on *task vectors*, the modifications made during fine-tuning, which is usually the information necessary to do well on a given task. Given a base model $\theta_{\text{base}}$ and a fine-tuned model $\theta_{\text{ft}}$, the corresponding task vector $\tau_{\text{task}}$ is defined as:

$$\tau_{\text{task}} = \theta_{\text{ft}} - \theta_{\text{base}}.$$

Task vectors provide an interpretable way to understand how fine-tuning adapts a model to a particular task. In the

| Category | Name | Size | Base Model |
|---|---|---|---|
| **VLMs** | LLaVA-NeXT (Liu et al., 2024) | 8B | LLaMA-series |
| | Idefics2-8B (Laurençon et al., 2024) | 8B | Mistral-series |
| | InternVL2 (Chen et al., 2024b) | 76B | LLaMA-series |
| | | **Domain** | **Base Model** |
| **Task Vectors** | Dart-Math (Tong et al., 2024) | Math Domain | LLaMA/Mistral-series |
| | MetaMath (Yu et al., 2024a) | Math Domain | LLaMA-series |
| | MAmmoTH-1 (Yue et al., 2024a) | Math Domain | LLaMA-series |
| | MAmmoTH-2 (Yue et al., 2024b) | General Domain | LLaMA/Mistral-series |
| | Magpie-v0.3 (Xu et al., 2024) | Math Domain | LLaMA-series |
| | Deepseek-R1-Distill (DeepSeek-AI et al., 2025) | Math Domain | LLaMA-series |

Table 1: Overview of Vision-Language Models (VLMs) and Task Vectors with the attributes -size, base models and domains.

context of VLMs, we define the adaptation of the language model component as:

$$\tau_{\text{vlm}} = \theta_{\text{vlm}} - \theta_{\text{base}}.$$

where $\tau_{\text{vlm}}$ captures the changes introduced when adapting the base LLM into the VLM. Similarly, for a reasoning-specialized LLM $\theta_{\text{reason}}$, we define its corresponding task vector:

$$\tau_{\text{reason}} = \theta_{\text{reason}} - \theta_{\text{base}}.$$

To enhance the reasoning ability of the VLM, we merge its language model with a strong reasoning-specialized LLM by linear merging:

$$\theta'_{\text{vlm}} = \theta_{\text{base}} + \lambda\tau_{\text{vlm}} + (1 - \lambda)\tau_{\text{reason}}.$$

Here, $\lambda$ determines the weight assigned to the VLM task vector, allowing us to control the balance between the original multi-modal capabilities of the VLM and the newly introduced reasoning strength from the LLM.

## 3. Experiment settings

In this section, we describe our experimental setup, detailing the selected models, datasets, and evaluation protocols used to evaluate the effectiveness of model merging and understand its internal workings.

**VLMs**  We span models in different sizes and base models to verify the generalization ability of merging. For VLMs, we use *LLaVA-Next-LLaMA3-8B* (Liu et al., 2024), *Idefics2-8B* (Laurençon et al., 2024), and *InternVL2-LLaMA3-76B* (Chen et al., 2024b) (Abbreviated as *LLaVA*, *Idefics*, and *InternVL* in our paper) ranging from 8B to 76B and including both Mistral-based and LLaMA-based models.

**Reasoning Task Vectors**  We span task vectors across different reasoning domains, beginning with mathematical reasoning tasks featured in Dart-Math (Tong et al.,

2024), which includes two variants: Dart-Uniform and Dart-Prop2diff. In this paper, we refer to the latter as Dart-Prop. Additionally, we examine MAmmoTH-1 (Yue et al., 2024a), Magpie-v0.3 (Xu et al., 2024), MetaMath (Yu et al., 2024a), and Deepseek-R1-Distill (DeepSeek-AI et al., 2025). Our scope further extends to broader reasoning task vectors obtained in MAmmoTH-2 (Yue et al., 2024b). The base models and task vectors are detailed in Table 1.

**Hyperparameters**  In our main analysis and experimental sections, we employ a linear merging strategy for all task vectors under the same hyperparameter settings to ensure a fair comparison. This approach assigns a weight of 0.9 to the textual component of *LLaVA-Next-LLaMA3-8B* and 0.1 to the reasoning task vector, where $\lambda = 0.9$. This parameter is tuned on MathVista based on Dart-Prop (Tong et al., 2024). We choose the best value from the range (0.8, 0.85, 0.9).

For our additional experiments analyzing the effects of merging across different base VLMs, we adjust the parameter within a range of 0.05 across the intervals (0.8, 0.85, 0.9) based on Dart-Prop on MathVista and apply the same hyperparameter across all benchmarks and other task vectors if they exist.

**Evaluation**  We evaluate the performance on a series of VLM benchmarks. We apply five benchmarks: Math-Vista (Lu et al., 2024), MathVerse (Zhang et al., 2025), MathVision (Wang et al., 2024a), Dynamath (Zou et al., 2024) and MMStar (Chen et al., 2024a).

Among these benchmarks, MathVista is a diverse benchmark that includes both math-related reasoning tasks and general visual question answering tasks. Each data sample in MathVista is meticulously annotated with meta-information such as source, task etc., allowing us to evaluate various aspects of improvement, such as identifying the specific scenarios where our method is most effective.

# 4. Can Model Merging Enhance VLM Capabilities?

In this section, we present model merging as a viable approach for enhancing inherent VLM capabilities, specifically exploring how merging with specialized models can augment core functionalities such as underdeveloped reasoning abilities in standard VLMs.

**Merging VLMs with math-specialized models consistently improves performance across all math benchmarks.** We posit that VLMs primarily rely on two fundamental capabilities: perception and chain-of-thought reasoning ability to solve the visual reasoning problems. Additionally, they also leverage the knowledge-recall ability to enhance their decision-making. Among these, chain-of-thought reasoning remains as a bottleneck in current VLMs (Zhang et al., 2024a), despite significant advances in this area by LLMs. This observation motivates our investigation into whether merging VLMs with math-specialized LLMs can enhance their inherent reasoning capabilities.

To explore this hypothesis, we use *LLaVA-Next-LLaMA3-8B*, a commonly employed VLM, as the base model, and integrate it with 5 state-of-the-art reasoning models (see task vectors in Table 1) that were specifically fine-tuned on reasoning tasks: Dart-Math (Tong et al., 2024) (Dart has two variants: Dart-Uniform and Dart-Prop2diff, abbreviated as Dart-Prop in our paper), MAmmoTH-1 (Yue et al., 2024a), MAmmoTH-2 (Yue et al., 2024b), and Magpie-v0.3 (Xu et al., 2024). For a fair comparison of the task vectors, we employ the linear merging strategy to all task vectors in the same hyper-parameter setting, which assigns a weight of 0.9 to the textual component of *LLaVA-Next-LLaMA3-8B* and 0.1 to the math task vector. The merged model is evaluated across five datasets (see §3), and the results are summarized in Table 2. As indicated by the green arrows, integrating VLMs with math-specialized models such as Dart (Tong et al., 2024) consistently improves performance over the baseline across all five mathematical datasets. Notably, on the MathVerse Benchmark in Text-Dominant evaluation mode, merging with Dart-Prop yields a 30% relative improvement over the baseline (a 6-point absolute increase). Additionally, it boosts performance on math-related VQA in MathVista by 3.5 absolute points. Whereas merging with a general-purpose reasoning LLM like MagPie (Xu et al., 2024) offers only modest gains.

Moreover, we extend our methods to other VLMs and task vectors. Table 3 presents results for Idefics2-8B (Yadav et al., 2024) and InternVL2-76B (Chen et al., 2024b) using task vectors available in the open-source community. For Idefics, MAmmoTH-1 achieves the best performance, improving accuracy by approximately 1.0 absolute points on average, while most reasoning task vectors fail to provide a

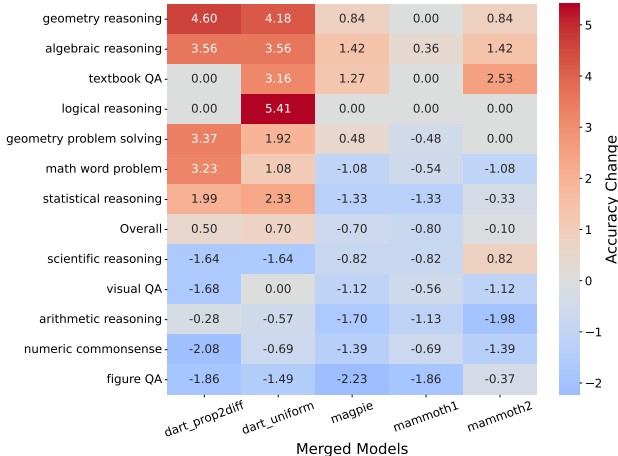

Figure 2: Accuracy changes after merging compared to the baseline. Generally, datasets directly requiring math-related and text-dominant capabilities, such as textbook QA and math word problem, exhibit clear improvements while domains requiring visual processing such as figure QA show performance degradation.

benefit. We attribute this to Idefics already being extensively fine-tuned on large-scale text-only data, including math SFT data, leading to a high degree of overlap with existing task vectors, which limits further improvements through merging. For InternVL2-76B, integrating Dart improves all benchmarks by approximately 1 absolute point, demonstrating that merging can also be beneficial for large-scale VLMs.

**Model merging exhibits minimal improvement or even decreased performance on vision-dominant tasks and general knowledge-centric tasks.** A closer analysis of performance across different subtasks reveals that on the MathVerse benchmarks, model merging yields higher performance gains for math-related and text-dominant samples, while vision-only questions show limited improvement (see Table 2). This phenomenon is also observed in the MathVista dataset (Figure 2), where we visualize performance changes relative to the baseline for each sub-domain dataset using a heatmap. **Notably, datasets that directly require math-related and text-dominant capabilities, such as geometric reasoning, algebraic reasoning and textbook question answering, exhibit consistent improvements.** However, domains requiring extensive visual processing (e.g., visual question answering and figure question answering) show slight performance degradation. Vision-only and vision-dominant tasks consist of questions embedded in figures, which require robust image perception to accurately recognize the question before employing knowledge-recall and reasoning to derive the answer. The inability of model merging to improve these tasks raises a critical question: if the bottleneck lies in image perception, then enhancing the

| Model | Method | MathVista | | | MathVerse Benchmarks | | | | | | MMStar | | DM | MV |
|---|---|---|---|---|---|---|---|---|---|---|---|---|---|---|
| | | All | General | Math | Overall | T-D | T-L | V-I | V-D | V-O | All | Math | | |
| **LLaVA** | Baseline | 37.4 | **51.7** | 25.4 | 20.1 | 25.9 | 20.8 | 21.1 | 16.5 | 16.0 | 43.8 | 30.0 | 22.7 | 13.8 |
| | +Dart-Uniform | **38.2** | 49.8 | 28.3 ↑2.9 | 23.6 ↑3.5 | **32.0** | **25.6** | 25.4 | 19.3 | **17.4** | 42.5 | 31.2 ↑1.2 | **24.5** ↑1.8 | 15.8 ↑2.0 |
| | +MAmmoTH-1 | 36.7 | 50.0 | 25.4 ↑0.0 | 21.1 ↑1.0 | 26.9 | 23.1 | 22.6 | 16.6 | 16.4 | **44.1** | 30.8 ↑0.8 | 22.6 ↓0.1 | 15.8 ↑2.0 |
| | +MAmmoTH-2 | 37.4 | 51.1 | 25.7 ↑0.3 | 20.6 ↑0.5 | 26.0 | 22.0 | 22.8 | 16.4 | 16.1 | 43.8 | 30.0 ↑0.0 | 22.5 ↓0.2 | 14.1 ↑0.3 |
| | +Magpie-v0.3 | 36.8 | 49.7 | 25.9 ↑0.5 | 20.7 ↑0.6 | 26.8 | 22.2 | 22.2 | 16.6 | 15.5 | **44.1** | 30.8 ↑0.8 | 22.7 ↑0.0 | **16.4** ↑2.6 |
| | +DeepSeek-R1-Distill | 38.1 | 51.1 | 27.0 ↑1.6 | 21.2 ↑1.1 | 28.4 | 22.7 | 22.5 | 17.3 | 15.1 | 43.7 | 33.2 ↑3.2 | 24.3 ↑1.6 | 15.1 ↑1.3 |
| | +Dart-prop | 38.0 | 48.7 | **28.9** ↑3.5 | **23.7** ↑3.6 | 30.7 | 24.8 | **25.5** | **19.8** | **17.4** | 43.6 | **33.6** ↑3.6 | **24.5** ↑1.8 | 14.8 ↑1.0 |

Table 2: The performance of *LLaVA-Next-LLaMA3-8B* model with merged task vectors across math-related Benchmarks: MathVista (All, General, and Math-related categories), MathVerse (Overall, Text-Dominant, Text-Lite, Vision-Integrated, Vision-Dominant, and Vision-Only categories), MMStar (All and Math split), DynaMath (annotated as DM), MathVision (annotated as MV). We include both variants of Dart (Tong et al., 2024) for comparison. We **bold** the highest value in each benchmark, and the gray row indicates the best task vectors on average.

| Model | Method | MathVista | | | MathVerse Benchmarks | | | | | | MMStar | | DM | MV |
|---|---|---|---|---|---|---|---|---|---|---|---|---|---|---|
| | | All | General | Math | Overall | T-D | T-L | V-I | V-D | V-O | All | Math | | |
| **Idefics** | Baseline | 51.8 | 57.0 | 47.4 | 19.4 | 24.4 | 21.3 | 20.7 | 19.7 | 11.0 | **49.5** | 39.6 | 21.8 | **17.1** |
| | +MetaMath | **53.2** | 57.8 | **49.3** ↑1.9 | 20.0 ↑0.6 | 25.3 | 22.3 | 21.1 | 18.7 | **12.4** | 48.1 | 39.2 ↓0.4 | 22.7 ↑0.9 | 11.8 ↓5.3 |
| | +Dart-Prop | 51.6 | 58.0 | 46.1 ↓1.3 | 20.0 ↑0.6 | 26.3 | 21.8 | **21.6** | 18.9 | 11.2 | 48.4 | 39.6 ↑0.0 | 22.7 ↑0.9 | 14.8 ↓2.3 |
| | +Dart-Uniform | 51.6 | 57.0 | 47.0 ↓0.4 | **20.5** ↑1.1 | **27.3** | 22.6 | 21.1 | 19.5 | 12.2 | 47.9 | 38.4 ↓1.2 | 22.7 ↑0.9 | 14.8 ↓2.3 |
| | +MAmmoTH-1 | 53.0 | **58.5** | 48.3 ↑0.9 | 20.4 ↑1.0 | 26.0 | 22.5 | 21.3 | **19.8** | 12.1 | 48.3 | **40.8** ↑1.2 | 23.2 ↑1.4 | 16.8 ↓0.3 |
| | +MAmmoTH-2 | 52.8 | 58.3 | 48.1 ↑0.7 | 18.4 ↓1.0 | 25.6 | **22.7** | 20.7 | 19.4 | **12.4** | 48.5 | 40.0 ↑0.4 | **24.0** ↑2.2 | 16.8 ↓0.3 |
| **InternVL** | Baseline | 65.6 | 67.0 | 64.4 | 43.1 | **54.1** | 47.5 | 44.8 | 43.8 | 25.3 | 67.3 | **75.2** | 38.7 | 23.7 |
| | +Dart-Uniform | **66.1** | **67.2** | **65.2** ↑0.8 | **44.3** ↑1.2 | 53.9 | **48.1** | **46.3** | **44.5** | **28.6** | **67.5** | 74.8 ↓0.4 | **39.6** ↑0.9 | **25.3** ↑1.6 |

Table 3: The performance of *Idefics2-8B* model and *InternVL2-LLaMA3-76B* model with merged task vector across math-related Benchmarks: MathVista (All, General, and Math-related categories), MathVerse (Overall, Text-Dominant, Text-Lite, Vision-Integrated, Vision-Dominant, and Vision-Only categories), MMStar (All and Math split), DynaMath (annotated as DM) and MathVision (annotated as MV). For benchmarks with Math subsets, only the Math score is included in the average score calculation. We **bold** the highest value in each benchmark, and the gray row indicates the best task vectors on average.

textual component through model merging may fail to yield performance gains.

**Merging with math models brings inference time scaling ability.** We hypothesize that the reasoning ability transferred to VLMs is primarily reflected in the improvement of chain-of-thought capabilities. To support this hypothesis, we analyze answer lengths before and after merging with Dart, highlighting significant shifts in task-specific behaviors. As shown in Figure 3, the performance improvement exhibits a nearly *linear relationship* with the increase in answer length, indicating that merging enables VLMs to scale inference time effectively. When looking closer at the specific tasks, chain-of-thought-intensive tasks such as "geometry problem solving", "geometry reasoning", and "algebraic reasoning" experienced a substantial increase in average prediction length, exceeding 250% of the original answer length. In contrast, changes in visual-intensive tasks like "figure question answering" and "visual question answering" were relatively modest, with nearly the same original answer

length, even showing a decrease in performance. These results suggest that merging with a math-focused model like Dart not only enhances the detail and depth of responses in reasoning-intensive tasks but also maintains efficiency and stability in perception-driven tasks, highlighting the adaptability of the merged framework across diverse domains. We show the details in Figure 9.

## 5. Merging as Interpretability Tool – Dive into the inner parameter space of LLaVA

In this section, we leverage model merging as an analytical tool to decompose and understand the internal mechanics of VLMs. By analyzing parameter modifications during the merging process, we aim to identify and isolate distinct parameter subspaces responsible for specific capabilities, such as visual perception and reasoning. We conduct a fine-grained analysis of LLaVA on the MathVista (Lu et al., 2024) dataset, which categorizes the examples into *"General VQA"* and *"Math VQA"*. We hypothesize that *"General*

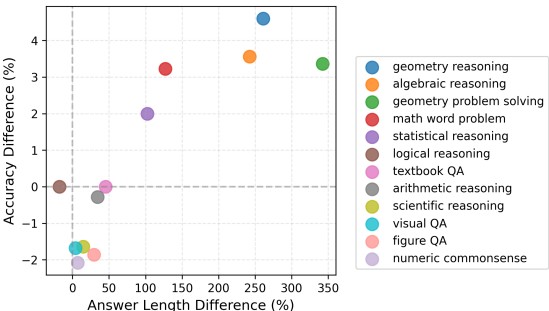

Figure 3: The relationship between answer length change (characters) and accuracy improvement after merging (The x-axis represents the relative change from the original answer, while the y-axis shows the absolute change in accuracy), classified by task and skills.

*VQA"* primarily evaluates *perception ability* and *knowledge recall*, while *"Math VQA"* assesses *perception ability* and *reasoning ability*.

**Mask Out**  We begin by using the **masking out** technique to assess the influence of each module. Specifically, for each layer, we replace the parameters of the MLP and attention modules with those from alternative modules (e.g., a uniform distribution or parameters from another model) to evaluate the absolute and relative impact of each module. Our hypothesis is that *a greater performance drop when masking a particular module indicates its higher importance for the task*, while a smaller drop suggests a more trivial effect on the task.

**Locate the perception ability in the parameter space**
We first analyze the impact of supervised fine-tuning (SFT) of LLaVA compared to the pretrain LLaMA, by progressively masking out LLaVA's parameters and replacing them with LLaMA's parameters, layer by layer for each module. Our hypothesis is that LLaVA's SFT training improves the model's perception ability, and we use the Masking Out technique to identify key regions where perception ability is located. ① and ③ at Figure 4 shows that, from a layer-wise perspective, masking out the early layers has a greater impact on general VQA tasks than masking the later layers, suggesting that **perception abilities are primarily located in the early layers**. We also observe that while general VQA performance declines after masking out, Math VQA performance improves consistently. This implies that LLaVA's SFT enhances perceptual abilities at the cost of reasoning capabilities, which motivates our efforts to improve reasoning ability in VLMs.

**Locate the chain-of-thought reasoning ability in the parameter space**  As shown in Table 2, LLaVA's per-

formance on math reasoning tasks significantly improves through merging. Furthermore, we demonstrate in §4 that this improvement stems from the infused inference-time scaling ability. To pinpoint where these infused chain-of-thought abilities reside in the parameter space, we masked the parameters of the "Dart-Merged LLaVA" using those from the original LLaVA model. ② and ④ at Figure 4 indicates that masking the later layers has a more pronounced impact on math-related tasks, suggesting that **the later five or more layers are crucial for math reasoning**. Moreover, merging with a math-focused model boosts math reasoning abilities while only minimally affecting general VQA performance. This implies that integrating a math model with small weights incurs only a slight trade-off in VQA capabilities while yielding significant gains for math-specific questions. Overall, these findings suggest that **perception and chain-of-thought reasoning abilities occupy distinct regions within the LLaVA parameter space and can be largely disentangled**.

**Expore the threshold of the absolute ability of LLaVA for each module**  In previous experiments, we masked individual modules using those from another model to comparatively evaluate specific capabilities. To precisely quantify the role of each module—such as attention or MLP layers—in VLMs for both general VQA and math VQA tasks, we replace each module's weights with a uniform distribution (i.e., assigning each parameter a value of $1/N$, where $N$ is the size of the weight matrix's first dimension). This substitution introduces significant noise, effectively disabling the module's functionality. By measuring the resulting performance drop, we can assess each layer's absolute contribution to the model's performance on both tasks.

Figure 5 shows the performance drops of disabling LLaVA (⑤ and ⑦) and Dart-Merged LLaVA (⑥ and ⑧). The first observation is that **1) early-to-middle layers are more crucial for both general and math-related tasks in the LLaVA model**, as evidenced by the significant drops, i.e., 25% absolute accuracy drop in general tasks and 10% in math-targeted VQA (highlighted in red gray square). This suggests that the early-to-middle layers play unique and indispensable roles in VLMs, this is intuitive since early layers handle perception, and accurately perceiving the image is a prerequisite for answering correctly while the later layers are less important and more robust to noise. Secondly, **math-targeted VQA shows a smaller performance drop than general VQA after masking out parameters, due to its inherently weak math reasoning ability.** When certain parameters are masked out, general visual question answering (VQA) tasks (left side of ⑤ and ⑦) exhibit a larger performance decrease than math-related questions (right side of ⑤ and ⑦). This can be explained by the LLaVA model's limited knowledge of math tasks and its inherent weak rea-

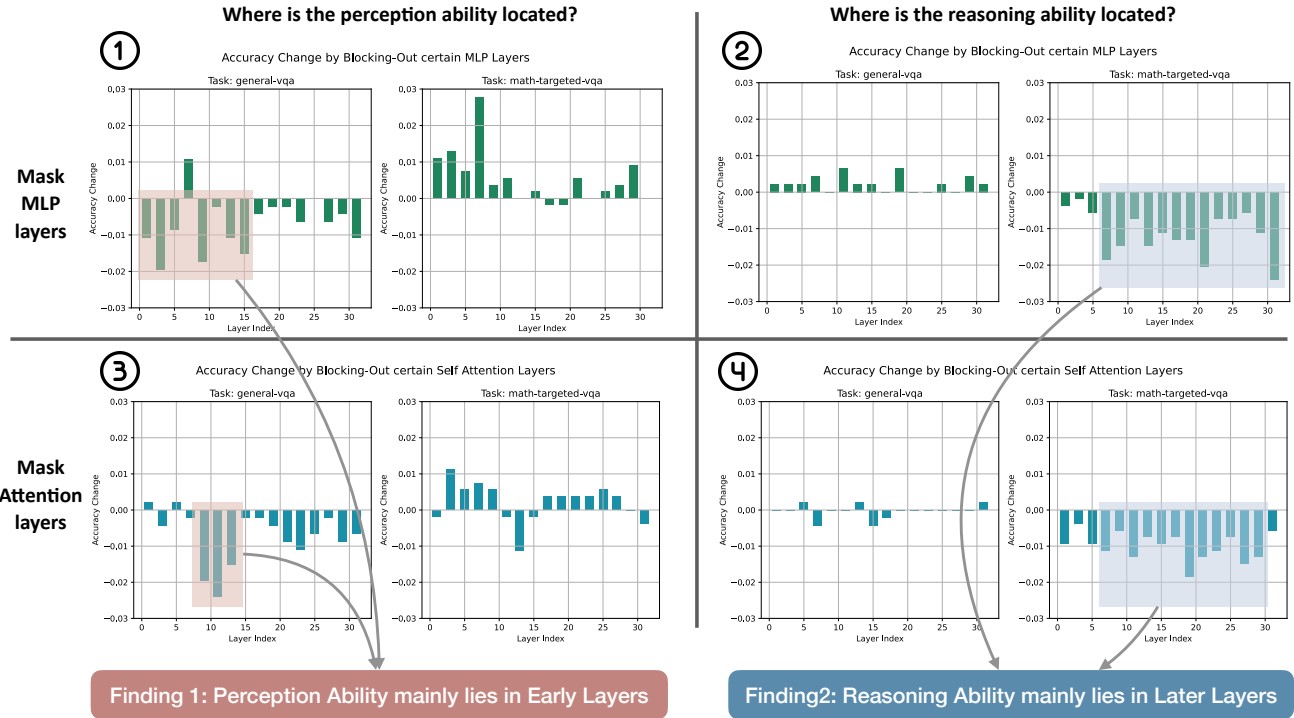

Figure 4: **LLaVA → LLaMA** (①and ③): the accuracy changes after replacing the parameters of each *MLP* (①) and *Attention* (③) layer of the LLaVA model with parameters from LLaMA. We find that masking out the early layers has a greater impact on general VQA tasks than masking the later layers, suggesting that **perception abilities gained from LLaVA-sft training are primarily located in the early layers**. **Dart-Merged LLaVA → LLaVA** (②and ④): the accuracy changes after replacing each *MLP* (②) and *Attention layer* (④) of the Dart-Merged LLaVA model to that of LLaVA. A significant drop in accuracy in math-targeted VQA tasks is observed **from 5 layer onwards.** (highlighted in blue), suggesting that the **reasoning ability** is mainly located on these layers.

soning abilities compared to the abilities demonstrated in VQA tasks (e.g., perception and world knowledge). In other words, the model's limited mathematical reasoning ability makes it less sensitive to parameter alterations, which also enhances our motivation to incorporate external reasoning ability into it.

**Merging with Reasoning Models Enhances Almost All Layers' Math Reasoning Ability** As shown in Figure 5, after merging with Dart (right side of ⑥and ⑧), we observe that nearly all layers—highlighted by the blue mask—drop more in math reasoning tasks compared to the base LLaVA (right side of ⑤and ⑦), where fewer layers are influenced. This suggests that reasoning ability has been successfully integrated into all layers without substantially affecting the layer distribution of perception ability. Several qualitative examples supporting this are presented in Figure 6. The first and third examples illustrate how the model better perceives key entities and makes decisions through chain-of-thought reasoning. However, for general VQA tasks, we also see reduced activation in the early layers, indicating a slight loss in world knowledge due to the model merging.

## 6. Related Work

**VLMs** Large Vision-Language Models (VLMs) consist of three main components: a visual encoder for processing images, such as CLIP (Radford et al., 2021) or SigLip (Zhai et al., 2023); a language model (e.g., a LLaMA model (Dubey et al., 2024) or a Mistral model (Jiang et al., 2023)) for processing textual inputs and image features to generate responses; and a projector, typically implemented as multilayer perceptron (MLP), to bridge the gap between the visual and language components. This module maps features from the visual space to the language space, facilitating interaction between the two modalities.

**Model Merging** Model merging offers a "free lunch" by repurposing fine-tuned models for downstream tasks (Wortsman et al., 2022; Ilharco et al., 2023; Zhang et al., 2023). It creates a new model through simple arithmetic on existing parameters, requiring no extra training or inference cost.

Several model merging techniques have been proposed to improve performance, such as calculating different weights for model parameters using data and internal model acti-

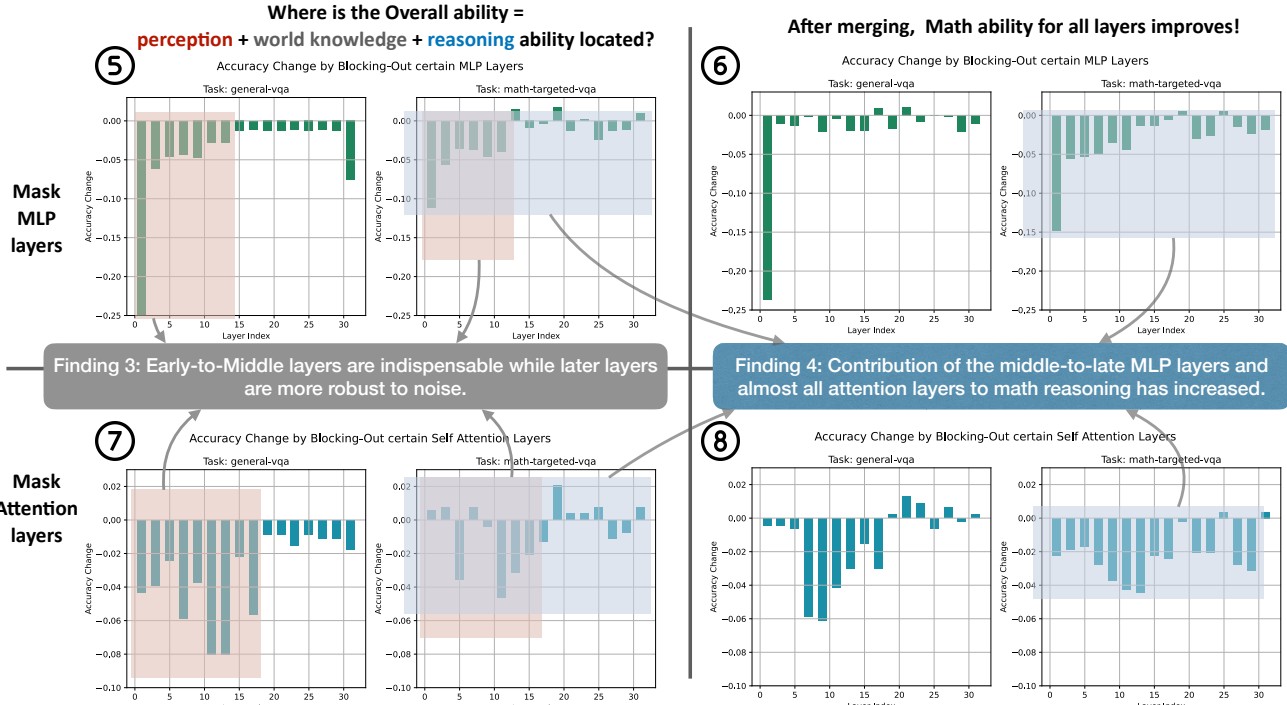

Figure 5: **LLaVA → 1/N** (⑤ and ⑦): the accuracy changes after replacing the parameters of each *MLP* (⑤) and *Attention* (⑦) layer of the LLaVA model with $\frac{1}{N}$, where $N$ is the first dimension of the weight matrix. The highlighted red area shows that early-to-middle layers are more crucial for both general and math-related tasks in the LLaVA model, as evidenced by the significant drops, i.e., 0.25 absolute accuracy drop in general tasks and 0.10 in math-targeted VQA. **Dart-Merged LLaVA → 1/N** (⑥ and ⑧): the accuracy changes after replacing the parameters of each *MLP* (⑥) and *Attention* (⑧) layer of the Dart-Merged LLaVA model with $\frac{1}{N}$. Comparing before and after merging when applied masking out, we observe a larger drop in accuracy in math-targeted VQA tasks **across all layers** (highlighted in blue), suggesting that the contribution of all most all layers to math reasoning has increased.

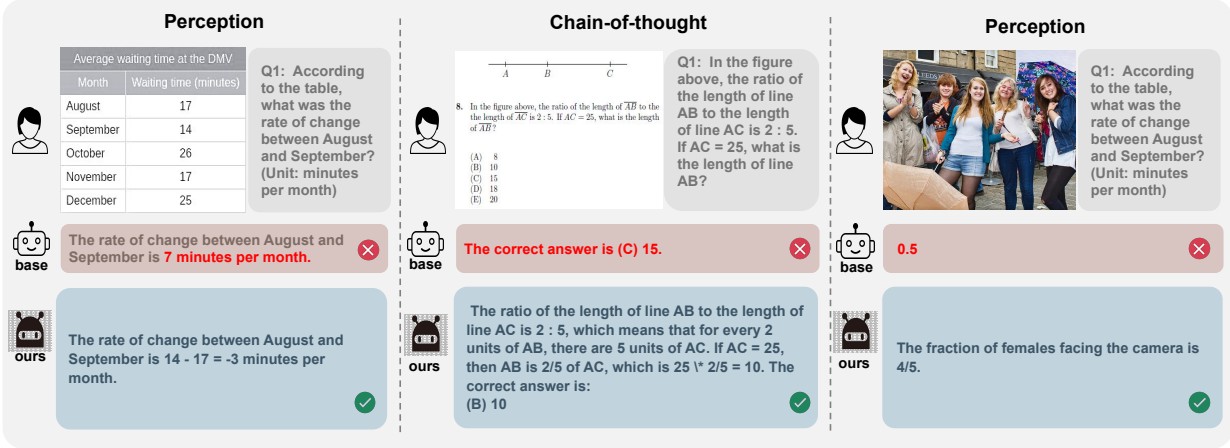

Figure 6: Qualitative study: three examples that can be fixed by merging with Dart (Tong et al., 2024).

vations (Matena & Raffel, 2022; Jin et al., 2023). Some approaches initially sparsify the models to reduce conflicts across different functions (Yadav et al., 2023; Yu et al., 2024b). Recently, Layer Swapping (Bandarkar et al., 2024)

was introduced, retains specific layers while merging others to enhance transfer learning. Despite their differences, simple averaging is often preferred for its simplicity and robustness. As model scales grow, performance gaps among

merging techniques shrink (Yadav et al., 2024), making linear merging a natural choice.

**Merging for VLMs**   Although much prior research has focused on merging vision models for tasks such as image recognition and scene understanding (Ilharco et al., 2023; Yang et al., 2023), the composition for VLMs remain insufficiently explored, particularly in terms of transferring specialized skills across modalities. Recently, REMEDY (Anonymous, 2025) proposes a practical merging recipe, which merges projector and front-layer modules of VLMs, focusing on multitasking and transfer in low-shot settings across various VQA types. Additionally, Sakana AI demonstrates the potential for multilingual capabilities in VLMs by transferring Japanese comprehension and generation abilities from an expert LLM to VLMs (Akiba et al., 2024). However, a key challenge lies in transferring reasoning skills across modalities. Our study addresses this by integrating the reasoning capabilities of LLMs into VLMs with a comprehensive analysis.

## 7. Conclusion

In this paper, we investigate the use of model merging methods to bridge the intelligence of cross-modalities, specifically transitioning from pure textual modality to vision-textual modalities. By incorporating various math-specific LLMs into different VLMs through model merging, we demonstrate that this approach effectively enhances the reasoning capabilities of VLMs. Our experimental results indicate an improvement of up to 12% in performance on math reasoning tasks compared to the baseline. And by employing model merging as an interpretability tool, we further unlock the parameter space of VLM. We find that for VLMs trained solely on image-text pairs, the abilities of perception and reasoning can be decomposed within the parameter space. Specifically, perception ability resides in the early layers, while reasoning ability is concentrated in the later layers. This decomposition further validates our experimental results.

## Impact Statement

This paper presents work whose goal is to advance the field of Machine Learning Interpretability. There are many potential societal consequences of our work, none of which we feel must be specifically highlighted here.

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

## A. The checkpoints used in experiments

We show all the checkpoints we use for experiments in Table 4.

| Category | Huggingface ckpt | Task Vectors | Huggingface ckpt of task vectors |
|---|---|---|---|
| **LLaVA-Next** | lmms-lab/llama3-llava-next-8b | MAmmoTH1 | EtashGuha/llama3-mammoth-dcft |
| | | MAmmoTH2 | TIGER-Lab/MAmmoTH2-8B |
| | | Magpie-v0.3 | Magpie-Align/Llama-3-8B-Magpie-Align-SFT-v0.3 |
| | | Dart-Uniform | hkust-nlp/dart-math-llama3-8b-uniform |
| | | Dart-Prop | hkust-nlp/dart-math-llama3-8b-prop2diff |
| | | DeepSeek-R1-Distill | deepseek-ai/DeepSeek-R1-Distill-Llama-8B |
| **Idefics2-8B** | HuggingFaceM4/idefics2-8b | MAmmoTH | TIGER-Lab/MAmmoTH-7B-Mistral |
| | | MAmmoTH2 | TIGER-Lab/MAmmoTH2-7B |
| | | MetaMath | meta-math/MetaMath-Mistral-7B |
| | | Dart-Uniform | hkust-nlp/dart-math-mistral-7b-uniform |
| | | Dart-Prop | hkust-nlp/dart-math-llama3-8b-prop2diff |
| **InternVL2** | OpenGVLab/InternVL2-Llama3-76B | Dart-Prop | hkust-nlp/dart-math-llama3-70b-prop2diff |

Table 4: All the huggingface checkpoints we use in our experiments

## B. More analysis for MathVista

MathVista includes various metadata, such as "Task" and "Task&Skills", allowing the dataset to be classified into subsets using different methods. We present MathVista's performance improvement after merging with Dart (Tong et al., 2024) across different "Tasks" (there are in total 5 tasks in MathVista) at Figure 7, along with the correlation between answer length and accuracy improvement for each task. We can see that two figures both show consistent pattern with findings at Section 4. Figure 7 suggests that the math-specialized task vectors primarily benefit math reasoning tasks, but may hinder performance on knowledge-intensive general VQA tasks. Figure 8 shows that merging with Dart enhances the model's inference-time scaling ability.

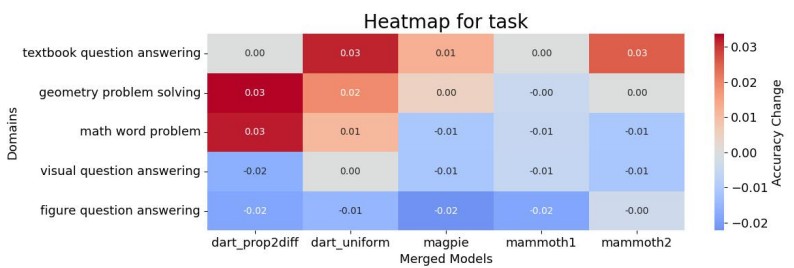

Figure 7: The accuracy difference after merging across several subtasks in MathVista for LLaVA.

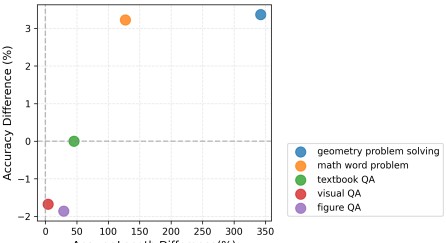

Figure 8: The relationship between answer length change and accuracy improvement after merging, classified by task.

## C. Different merging methods perform generally comparable

We are interested in whether different merging methods affect the performance of reasoning ability transfer. To investigate this, we employ TIES (Yadav et al., 2023), Dare merging (Dare-TIES and Dare-Linear) (Yu et al., 2024b), and Layer Swapping (Bandarkar et al., 2024), which are popular merging methods in practice, to compare their performance with that of linear merging. We adopt the hyperparameter search strategy as linear merging, parameterized by $(\alpha_1, \alpha_2)$, where $\alpha_1$ is tuned for the VLM vector and $\alpha_2$ for the Math LLM task vector, to obtain the most comparable checkpoints on benchmarks emphasizing visual and textual reasoning.

As shown in Figure 10, we present results of each configuration for TIES merging on both the visual- and text-dominant tasks,

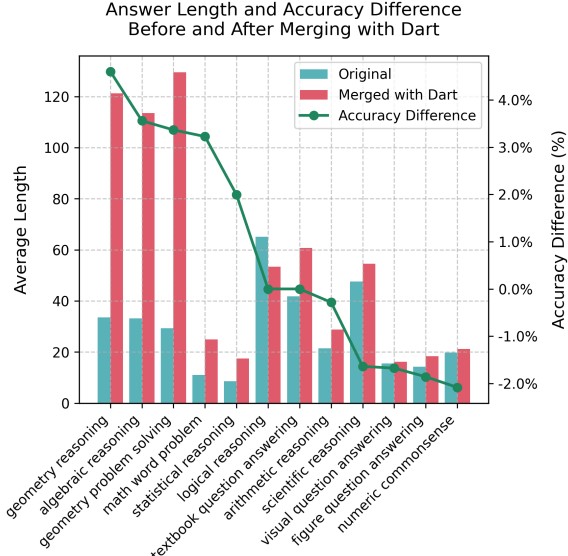

Figure 9: The relationship between answer length change and accuracy improvement after merging, classified by task.

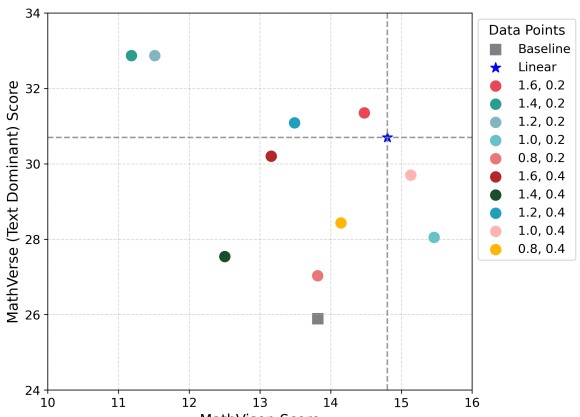

Figure 10: Performance comparison of TIES and linear merging methods across MathVerse-Text-Dominant) and MathVision benchmarks. Baseline presents the result of *LLaVA-Next-LLaMA3-8B*. Linear shows the result of merging Dart-prop vector with hyperparameters of $(0.9, 0.1)$. Performance reached by TIES merging with the same two task vectors are labeled as the hyperparameter pair used.

| Method | Weights | **MathVision** | **MathVerse-Text-Dominant** |
|---|---|---|---|
| Baseline | - | 13.8 | 25.9 |
| Linear | (0.9, 0.1) | 14.8 | 30.7 |
| TIES | (1.6, 0.2) | 14.5 | 31.4 |
| Dare-TIES | (1.0, 0.2) | 17.8 | 22.7 |
| Dare-Linear | (1.2, 0.2) | 17.4 | 21.7 |
| Swap5Layers | (0.9, 0.1) | 15.1 | 29.6 |

Table 5: Performance comparison of different merging methods on MathVision and MathVerse-Text-Dominant datasets.

MathVision and MathVerse-Text-Dominant, accordingly. Compared to the performance yielded by linear merging, most combinations of TIES trade off gains in one domain against losses in the other. Additionally, the constraint of $\alpha_1 + \alpha_2 = 1$ is commonly dropped in practice because of sparsification during TIES merging process. As a result, TIES merging requires more hyperparameter tuning to achieve comparable performance in a expanded searching space.

We also expand our experiments on the other methods and results shown in Table 5. The results performs comparably to TIES merging, consistent with our finding that different merging methods yield similar performance, with none significantly outperforming simple averaging. This supports our choice to adopt the linear merging method. This finding is also consistent with the previous finding in Yadav et al. (2024) that different merging methods tend to exhibit similar behaviors at larger scales. Given this, we chose not to focus extensively on exploring alternative merging methods but instead to explore more about the interpretability and composition of the model's internal abilities.

## D. Additional results demonstrating generalization from logical to mathematical reasoning

To investigate whether reasoning skills from other domains, such as logical reasoning, can generalize to multi-modal math reasoning, we fine-tune LLaMA3-8B on LogiCoT (Liu et al., 2023)—a logical chain-of-thought dataset—and merge it with LLaVA. As shown in Table 6, this logic-focused model improves performance on math reasoning tasks, suggesting generalization capabilities across the reasoning domains and highlighting the potential for transferring reasoning skills from textual to multi-modal settings.

| Model | Method | MathVista | | | MathVerse Benchmarks | | | | | | MV |
|---|---|---|---|---|---|---|---|---|---|---|---|
| | | All | General | Math | Overall | T–D | T–L | V–I | V–D | V–O | |
| LLaVA | Baseline | 37.4 | 51.7 | 25.4 | 20.1 | 25.9 | 20.8 | 21.1 | 16.5 | 16.0 | 13.8 |
| | +logic | 37.0 ↓0.4 | 49.1 ↓2.6 | 26.7 ↑1.3 | 23.5 ↑3.4 | 30.5 ↑4.6 | 25.0 ↑4.2 | 26.4 ↑5.3 | 20.3 ↑3.8 | 15.1 ↓0.9 | 11.2 ↓2.6 |

Table 6: Performance (%) of **LLaVA** before and after merging with the LLM fine-tuned on logical training data (Liu et al., 2023) on MathVista, MathVerse and MathVision benchmarks. Arrows denote absolute change from the baseline.

| Model | Method | MathVista | | | MathVerse Benchmarks | | | | | | MMStar | | DM | MV |
|---|---|---|---|---|---|---|---|---|---|---|---|---|---|---|
| | | All | General | Math | Overall | T–D | T–L | V–I | V–D | V–O | All | Math | | |
| Qwen2-VL | Baseline | 61.2 | 69.6 | 54.1 | 31.8 | 35.9 | 31.4 | 31.5 | 33.1 | 26.9 | 59.9 | 59.2 | 34.4 | 21.1 |
| | +Qwen2-Math | 60.2 | 68.0 | 53.5 ↓0.6 | 31.9 ↑0.1 | 37.1 | 31.7 | 31.5 | 32.5 | 26.7 | 59.5 | 58.4 ↓0.8 | 35.0 ↑0.6 | 21.7 ↑0.6 |

Table 7: Performance (%) of **Qwen2-VL** before and after augmenting with Qwen2-Math on MathVista, MathVerse Benchmarks, MMStar, DM and MV. Arrows denote absolute change from the baseline.

# E. More results on Qwen2-VL

To assess the generality of this approach, we further extend our experiments to a stronger vision–language model, Qwen2-VL. In Table 7, we present the results of merging Qwen2-VL-7B-Instruct with Qwen2-Math-7B. The merged model shows significant improvements on MathVerse, DynaMath and MathVision, but a performance drop on MathVista and MMStar. This uneven pattern mirrors our observations with Idefics and InternVL, suggesting that VLMs already pretrained on mathematics-focused text corpora—as is common in many state-of-the-art models today—derive less benefit from integration with specialized reasoning modules (Laurençon et al., 2024).

# F. More results on MM-Math

| Model | Method | MM-Math |
|---|---|---|
| LLaVA | Baseline | 0.61 |
| | +Dart-prop | 0.71 ↑0.10 |
| | +Dart-uniform | 0.86 ↑0.25 |
| | +MAmmoTH-1 | 0.68 ↑0.07 |
| | +MAmmoTH-2 | **1.46** ↑0.85 |
| | +Magpie-v0.3 | 1.30 ↑0.69 |
| | +DeepSeek-R1 | 0.27 ↓0.34 |
| | +Dart-keep-5layers | 1.05 ↑0.44 |
| Idefics | Baseline | 4.00 |
| | +MetaMath | **4.68** ↑0.68 |
| | +Dart-prop | 2.63 ↓1.37 |
| | +Dart-uniform | 2.73 ↓1.27 |
| | +MAmmoTH-1 | 4.03 ↑0.03 |
| | +MAmmoTH-2 | 3.80 ↓0.20 |
| InternVL | Baseline | 22.70 |
| | +Dart-uniform | **22.80** ↑0.10 |

Table 8: Addition experiment result on MM-Math.

In Table 8, we provide detailed results of our method applied to the MM-Math (Sun et al., 2024) benchmark, presenting general improvement brought by model merging in visual reasoning.

## G. Significance Test

To evaluate whether the improvements from merging are significant, we conduct a significance test and mark in Table **??**. Merging general reasoning models like Mammoth2 and Magpie does not yield statistically significant enhancements, exhibiting a pattern of marginal accuracy gains, as shown in Table 2 of our paper. In contrast, merging math-focused LLMs, such as Dart-Prop, results in statistically significant improvements. This supports our conclusion that merging with math-related models offers the greatest advantage.

| Model | Method | MathVista | MathVerse Benchmarks | | | | | | MMStar | DM |
|---|---|---|---|---|---|---|---|---|---|---|
| | | Math | Overall | T-D | T-L | V-I | V-D | V-O | Math | |
| | Baseline | 25.4 | 20.1 | 25.9 | 20.8 | 21.1 | 16.5 | 16.0 | 30.0 | 22.7 |
| | +Dart-Uniform | 28.3 ↑2.9 | 23.6 ↑3.5 | **32.0**$^*$ | **25.6**$^*$ | 25.4$^*$ | 19.3$^*$ | **17.4** | 31.6 ↑1.2 | **24.5** ↑**1.8** |
| | +MAmmoTH-2 | 25.7 ↑0.3 | 20.6 ↑0.5 | 26.0 | 22.0 | 22.1$^*$ | 16.4 | 16.1 | 30.0 ↑0.0 | 22.5 ↓0.2 |
| LLaVA | +Magpie-v0.3 | 25.9 ↑0.5 | 20.7 ↑0.6 | 26.8 | 22.2 | 22.6 | 16.2 | 15.5 | 30.8 ↑0.8 | 22.7 ↑0.0 |
| | +Dart-prop | **28.9** ↑**3.5**$^*$ | **23.7** ↑**3.6**$^*$ | 30.7$^*$ | 24.8$^*$ | **25.5**$^*$ | **19.5**$^*$ | **17.4**$^*$ | **33.6** ↑**3.6**$^*$ | **24.5** ↑**1.8**$^*$ |

Table 9: Significant test results across the models and datasets. $^*$ marks statistically significant improvements ($p < 0.05$).

