# OpenReview forum: "Bring Reason to Vision: Understanding Perception and Reasoning through Model Merging"
_ICML.cc/2025/Conference — ICML 2025 poster_

### Official Review · Reviewer_xnvF · 2025-03-11

**Overall Recommendation:** 3

**Summary:**

This is an experimental paper. This paper enhances VLMs with reasoning capabilities using Reasoning LLMs. Specifically, this paper integrates reasoning capability of Reasoning LLMs into VLMs through linear merging. Extensive experiments demonstrates the effectiveness of this simple model merging method on math reasoning tasks. Additionally, this paper experimentally disentangles perceptual and reasoning abilities within the model parameter space.

**Claims And Evidence:**

Yes.

**Essential References Not Discussed:**

No.

**Experimental Designs Or Analyses:**

Recently, there are many reasoning LLMs based on Qwen-series, I think the author should consider including this series in experimental designs to further valid their claims.

**Methods And Evaluation Criteria:**

Yes.

**Other Comments Or Suggestions:**

No

**Other Strengths And Weaknesses:**

1. To valid the effectiveness of merging model to enable reasoning capability of VLMs, more comprehensive experiments should be conducted. Like Qwen-based model, Results of InternVL in other benchmark.
2. The technical contribution is minimal. I would not be inclined to reject a paper based solely on the method. But the workload of experimental analysis and discovery in this paper is not enough to support it to be accepted by a top-tier conference. There is still a lot to do after getting the analysis and discovery. For examples, as the author found that the perception ability primarily located in the early layers and the reasoning ability mainly lied in later layers, why not try a dynamic model merging method considering this finding?

**Questions For Authors:**

1. Can the author try to include Qwen-based model to futher valid the effectiveness of model merging?

2. Why not report the results of InternVL on other benchmarks considering that LLaVA and Idefics report them. Can the author also reports them to make the claim more confidential.

3. Based on the finding in Sec. 5, Did the author try to only merge reasoning llm in the later layers? would this allow VLMs to achieve a good balance between perception and reasoning? For multimodal math reasoning benchmarks, there is no doubt that VLMs also needs a strong perception to understand the image input.

**Relation To Broader Scientific Literature:**

Methods in this paper is widely used. But as far as I know, I haven't seen similar findings of this paper in other literature.

**Theoretical Claims:**

They experimentally proof their claim in Sec. 4 and Sec. 5.

---

> ### Author Rebuttal · Authors · 2025-04-01
>
> Thanks for the comments! We address the concerns below.
>
> W1:
> >The workload of experimental analysis and discovery in this paper is not enough…For examples…why not try a dynamic model merging method considering this finding?
>
> We sincerely appreciate your feedback on our work. But we respectfully disagree with the comments on “insufficient workload” for the following reasons:
> - A Simple Method Does Not Mean Less Work: We actually tried it in our early experiments but it worked similarly to the simple linear merging method [see response to Q3]. Given no clear performance gain, we opted for the simpler solution.
>  - We want to emphasize here again that our goals are:
>     - **To see whether the perception and reasoning abilities can be infused by model merging**. We incorporate three VLMs of varying architectures and sizes, along with six reasoning models across diverse domains, and evaluate on five multimodal reasoning benchmarks.This evaluation process accruing costs of around $1,000.
>     - **To understand how perception and reasoning abilities are distributed in parameter space** within a VLM. To achieve this: In Figures 4 & 5, we conducted masking experiments on each even layer (MLP/Attention) under various configurations (e.g., masking LLaVA to LLaMA or random noise), evaluating each independently—resulting in 128 runs in total. In addition, we conducted 4–5 such preliminary experiments. We use GPT-4 (via VLMEvalKit) to evaluate model outputs, accruing costs of around $3,000.
>      - Although **GPU hours and monetary costs** do not directly equate to the quality of the work, these data points **do illustrate one aspect of the hidden workload** we have put into this work.
>
> Besides, we also add results based on Qwen model [see response to Q1], hopefully these added results can help recognize the workload of this paper.
>
> Q1/2:
> >include Qwen-based model.., report the results of InternVL on other benchmarks considering that LLaVA and Idefics .
>
> Thank you for the advice! We expand our experiments for Qwen-based model here. Specifically, we merge Qwen2-VL-7B-Instruct with Qwen2-Math-7B, the results are:
> |**Model**|**Method**|**MathVista**|||**MathVerseBenchmarks** ||||||**MMStar**||**DM** |**MV**|
> |-|-|:-:|:-:|:-:|:-:|:-:|:-:|:-:|:-:|:-:|:-:|:-:|:-:|:-:|
> |||All|General|Math|Overall|T-D|T-L|V-I|V-D|V-O|All|Math|||
> |Qwen2-VL|Baseline|61.2|69.6|54.1|31.8|35.9|31.4|31.5|33.1|26.9|59.9|59.2|34.4|21.1|
> ||+Qwen2-Math|60.2|68.0|53.5↓0.6|31.9↑0.1|37.1|31.7|31.5|32.5|26.7|59.5|58.4↓0.8 |35.0↑0.6|21.7↑0.6|
>
> From the results, we see that merging benefits certain benchmarks like MathVerse, DynaMath and MathVision. However, there is a decline in performance for benchmarks MathVista and MMStar. The unstable increase pattern is similar to what we observe with Idefics and InternVL, indicating that VLMs that have undergone math-related pure text training  (as many state-of-the-art models do today [1]) benefit less from merging with reasoning models.
>
> As for the large-scale model InternVL-76B, we have already evaluated it on the same benchmarks, as shown in the InternVL row of Figure 3 in our paper.
>
> Q3:
> > Did the author try to only merge reasoning llm in the later layers? would this allow VLMs to achieve a good balance between perception and reasoning?
>
> Thank you for the great suggestion of trying a dynamic model merging method! Like we have briefly discussed this in W1 response, we did test a dynamic approach based on our layer-wise insights.Inspired by the observation that perceptual abilities lie in early layers, we merge only the later layers to preserve them.
>
> |**Model**|**Method**|**MathVista**|||**MathVerseBenchmarks**||||||**MMStar**||**DM**|**MV**|
> |-|-|:-:|:-:|:-:|:-:|:-:|:-:|:-:|:-:|:-:|:-:|:-:|:-:|:-:|
> |||All|General|Math|Overall|T-D|T-L|V-I|V-D|V-O|All|Math|||
> |LLaVA|Baseline|37.4|51.7|25.4|20.1|25.9|20.8|21.1|16.5|16.0|43.8|30.0|22.8|11.8|
> ||+Dart-prop|38.0|48.7|28.9↑3.5|23.7↑3.6|30.7|24.8|25.5|19.8|17.4|43.6|33.6↑3.6|24.5↑1.7|14.8↑3.0|
> ||+keep5layers|37.9|48.5|28.9↑3.5|22.0↑1.9|29.6|23.2|23.2|17.8|16.4|43.8|30.8↑0.8|23.7↑0.9|15.1↑3.3|
>
> Although the results show clear gains over the baseline, they are only on par with linear merging. Given no clear performance gain, we opted for the simpler solution. One possible reason is that interpretation-based analysis is inherently subjective and only captures coarse-grained trends, often with substantial noise. For instance, while later layers may show stronger reasoning, earlier layers still play a role. Thus, merging only the later layers in reasoning-focused LLMs may not yield performance gains. Decomposing the two abilities may enable successful merging but doesn’t necessarily yield a better merging method. This also consists with previous findings from [2] that **different merging methods behave very similarly at larger scales**.
>
> [1] What matters when building vision-language models? NeurIPS 2024.
> [2] What matters for model merging at scale? Prateek et al. 2024

---

> > ### Comment · Reviewer_xnvF · 2025-04-02
> >
> > Further Concern: I still think the method of this paper is to apply method A (linear merging) to field B (multimodal reasoning) and it can work directly. No difficulties were encountered, and there was no need to consider how to solve them during applying A to B. Can the author further explain this point?

---

> > > ### Author Response · Authors · 2025-04-03
> > >
> > > Thanks for your prompt reply. We truly appreciate your feedback and will revise the paper accordingly to better highlight its contributions.
> > >
> > > Firstly, we would like to respectfully clarify a **misunderstanding**:
> > >
> > >  - Our paper is **not just a method paper**. Instead, we aim to **use model merging as a lens to understand** *how perception and reasoning ability are encoded and interact* within vision-language models.
> > >  - The merging method itself is a means to an end—a byproduct of our broader interpretability analysis.
> > >
> > > We argue that understanding the internal dynamics of perception and reasoning from a mechanistic interpretability perspective is also a valuable and underexplored scientific direction.
> > >
> > > Secondly, we want to summarize our **core contributions** here, which go beyond simply applying an existing method:
> > > 1. **A novel approach for mechanistic interpretability**: We are the first, to our knowledge, to use model merging as a tool to investigate where and how perception and reasoning are encoded in a large vision-language model.
> > > 2. **Non-trivial and practical empirical observation**: As pointed out in this feedback, we show that *simple linear merging can successfully merge perception and reasoning abilities across modalities*. However, we want to emphasize that the information is **not straightforward and trivial** as it may seem to be. For example, **prior work [1]** shows that naively interpolating two neural networks with entirely disjoint optimization trajectories can lead to a **catastrophic drop** in accuracy. In fact, this drop is quite common in model merging and is potentially due to conflicting parameter updates [2,3,4]. Thus, our finding that linear merging works across modalities for combining reasoning and perception abilities is of value for both the research and practitioner communities.
> > > 3. **Successful localization of distinct functional subspaces**: In Section 5, we design experiments that selectively mask parameters (e.g., replacing original Llava parameters with Llama’s weights or random noise) to localize perception and reasoning layers before and after merging, and reveal how the two abilities interact or stay disentangled after merging. Our results suggest that **perception and reasoning lie in largely orthogonal subspaces** in parameter space, **allowing for modular and effective merging**.
> > >
> > > In short, we are not merely proposing to “apply method A to field B”. Instead, we ask: **Can we infuse perception and reasoning abilities from different modalities? And how does it work?** And we provide both tools and insights to answer that question.
> > >
> > > We hope the above clarification helps you reconsider the contribution of our work, from both empirical and interpretability perspectives. Thank you again for your time and thoughtful feedback.
> > >
> > > [1] What is being transferred in transfer learning? Behnam Neyshabur. et al. ICLR 2020.
> > > [2] What matters for model merging at scale? Prateek et al. 2024.
> > > [3] TIES-Merging: Resolving Interference When Merging Models. Prateek Yadav.NeurIPS 2023.
> > > [4] Language Models are Super Mario: Absorbing Abilities from Homologous Models as a Free Lunch. Le Yu et al. ICML 2024.

---

### Official Review · Reviewer_V8q8 · 2025-03-12

**Overall Recommendation:** 3

**Summary:**

This paper proposes merging models across modalities to enable the incorporation of the reasoning capabilities of LLMs into VLMs. Effectiveness of this training-free recipe is verified via extensive experiments. It also evaluates VLMs in different sizes to verify the generalization ability of merging. Additionally, this work provide interesting insights about perception and reasoning abilities across different layers by investigating into merged VLMs.

"## update after rebuttal
While I appreciate the authors’ efforts to address most of my concerns, their response to my questions about leveraging the finding—that perception and reasoning abilities lie in different areas of MLLMs—for improved merging performance is not insightful enough. It's claimed that different merging methods behave very similarly at larger scales. Consequently, I maintain my original score.

**Claims And Evidence:**

Yes, via extensive experiments and case study of LLaVA.

**Essential References Not Discussed:**

No

**Experimental Designs Or Analyses:**

Yes, I've checked the main experiments and analysis of merged LLaVA.

**Methods And Evaluation Criteria:**

Yes, the reasoning ability is measured by math reasoning benchmarks, while perception ability is measured by general VQA benchmarks.

**Other Comments Or Suggestions:**

See Strengths And Weaknesses

**Other Strengths And Weaknesses:**

## Strength:
1. The problem addressed is timely and interesting, it might provide a new perspective for inference time scaling in VLMs.
2. It conducted extensive experiments on different size VLMs, demonstrating noticeable improvements and good generalization ability.
3. The analysis focusing on merged LLaVA provide insights about perception and reasoning patterns across different layers.

## Weakness:
1. Lack of baselines of other merging methods. Only TIES is provided in the appendix.
2. The investigation about perception and reasoning pattern across layers only considers LLaVA. It would be more interesting to also include other model families or larger models, such as Idefics2-8B, InternVL2-LLaMA3-76B.
3. Lack of significance test since many of the performance improvements in table 2 are within a 1% margin.

**Questions For Authors:**

I'm interested in how to leverage the findings that perception and reasoning abilities lie in different areas of VLMs, so that we can  achieve better merging performance.

**Relation To Broader Scientific Literature:**

The main contribution of this work is to apply a training-free recipe to merge VLMs with reasoning expert models. It's more a combinational innovation because the linear merge recipe was proposed by prior work.

**Theoretical Claims:**

There's no theoretical claims in this paper.

---

> ### Author Rebuttal · Authors · 2025-04-01
>
> We appreciate your thorough review and detailed comments! Your suggestions will be helpful in improving the paper.
>
> Q1:
> > Lack of baselines of other merging methods. Only TIES is provided in the appendix.
>
> We appreciate the suggestion and have expanded our experiments to evaluate DARE merging [3] (Dare-TIES and Dare-Linear), another SOTA merging method to explore additional merging techniques. We follow the hyperparameter search strategy same as TIES, parameterized by ($\alpha_1$, $\alpha_2$), where $\alpha_1$ is tuned for the VLM and $\alpha_2$ for the Math LLM, in order to obtain the most comparable checkpoints on benchmarks that emphasize vision and textual reasoning. The results are shown below.
>
> ||**MathVision**|**MathVerse-Text-Dominant**|
> |:-:|:-:|:-:|
> |Baseline|13.8|25.9|
> |Linear|14.8|30.7|
> |TIES(1,0.4)|15.1|29.7|
> |TIES(1.6,0.2)|14.5|31.4|
> |Dare-TIES(0.8,0.2)|17.8|22.7|
> |Dare-Linear(0.8,0.2)|17.4|21.7|
>
> The results show that DARE performs comparably to linear merging and TIES, consistent with our finding that different merging methods yield similar performance (line 657), with none significantly outperforming simple averaging. This supports our choice to adopt the linear merging method in our paper. This finding is also consistent with the previous finding in [2] that different merging methods tend to exhibit similar behaviors at larger scales. Given this, we chose not to focus extensively on exploring alternative merging methods but instead to explore more about the interpretability and composition of the model’s internal abilities.
>
> Q2:
> >It would be more interesting to also include other model families or larger models.
>
> We focused on LLaVA for our interpretability experiments mainly due to resource limits. Specifically, we conducted masking-out experiments on each even layer (for MLP/Attention, respectively) across four configurations (e.g., masking LLaVA to LLaMA). These experiments, presented in Figures 4 and 5, involved a total of 128 runs. We use GPT-4 to evaluate these runs (by VLMEvalKit), which incurred a substantial cost of approximately $3,000. Due to the high computational cost, our study currently focuses on LLaVA. However, we recognize the value of extending this analysis to other model families and leave it as promising future work.
>
> Q3:
> >Lack of significance test…
>
> Thank you for your valuable feedback! We acknowledge that some of the performance improvements in Table 2 fall within a 1% margin. We conduct t-tests to compare the results of the merged model with the base LLaVA. The resulting p-values are presented in the table below, with statistically significant results (p < 0.05) shown in bold. Merging general reasoning models such as Mammoth2 and Magpie does not lead to statistically significant improvements, showing a similar pattern of marginal accuracy gains as shown in Table 1 of our paper. In contrast, merging Math LLMs, such as Dart-Prop, leads to statistically significant improvements. This aligns with our conclusion in the paper (line 204) that merging with math-related models provides the greatest benefit.
>
> |**Model**|**Method**|**MathVista**|**MathVerse**|||||**MMStar**|**DM**|
> |-|-|:-:|:-:|:-:|:-:|:-:|:-:|:-:|:-:|
> |||Math|T-D|T-L|V-I|V-D|V-O|Math||
> |LLaVA|+Dart-Uniform|0.15|**0.00**|**0.00**|**0.00**|**0.02**|0.46|0.56|0.21|
> ||+MAmmoTH-2|0.77|0.58|0.30|**0.04**|0.70|0.35|1.00|1.00|0.82|1.00|
> ||+Magpie-v0.3|0.67|1.00|0.06|0.07|0.84|0.62|0.56|0.96|
> ||+Dart-prop|0.06|**0.00**|**0.01**|**0.00**|**0.01**|**0.02**|**0.05**|**0.00**|
>
> Q4:
> > I'm interested in how to leverage the findings that perception and reasoning abilities lie in different areas of VLMs, so that we can achieve better merging performance.
>
> During our exploration, we wondered whether insights from the decomposition of perception and reasoning abilities could be leveraged to develop better merging methods, similar to the approach proposed by [1]. To investigate this, we experimented by keeping the early layers unchanged to preserve perceptual ability while only merging the later layers to enhance reasoning ability (see Q3 to Reviewer xnvF for details). Our results indicate that this approach does not yield any performance gains.  Given no clear performance gain, we opted for the simpler solution. Our finding is also consistent with previous findings from [2] that **different merging methods behave very similarly at larger scales**.
>
> [1] Layer Swapping for Zero-Shot Cross-Lingual Transfer in Large Language Models, Lucas et al. ICLR 2025.
> [2] What Matters for Model Merging at Scale?. Yadav et al. arXiv 2024.
> [3] Language models are super mario: Absorbing abilities from homologous models as a free lunch. Yu et al.ICML. 2024.

---

> > ### Comment · Reviewer_V8q8 · 2025-04-03
> >
> > Thank you for the detailed explanations. Most of my concers are addressed.
> >
> > Regarding Q3, could you please provide a revised version of Table 2 to indicate p < 0.05 with underscore?

---

> > > ### Author Response · Authors · 2025-04-03
> > >
> > > Thank you for your response! We present the updated Table 2 with significance tests included below (results with * indicate p < 0.05). Our conclusion remains consistent: merging with math-related models such as Dart yields greater improvements in reasoning ability compared to general reasoning models, as supported by the significance test results. We will update the Table in our paper accordingly and thank you for your valuable suggestion!
> > >
> > > |**Model**|**Method**|**MathVista**||**MathVerse Benchmarks**||||||**MMStar**||**DM**|**MV**|
> > > |-|-|:-:|:-:|:-:|:-:|:-:|:-:|:-:|:-:|:-:|:-:|:-:|:-:|
> > > |||All|Math|Overall|T-D|T-L|V-I|V-D|V-O|All|Math|||
> > > |LLaVA|Baseline|37.4|25.4|20.1|25.9|20.8|21.1|16.5|16.0|43.8|30.0|22.7|13.8|
> > > ||+Dart-Uniform|**38.2**|28.3 ↑2.9|23.6 ↑3.5|**32.0***|**25.6***|25.4*|19.3*|**17.4**|42.5|31.6 ↑1.2|**24.5 ↑1.8**|15.8 ↑2.0|
> > > ||+MAmmoTH-1|36.7|25.4 ↑0.0|21.1 ↑1.0|26.9|23.1*|22.6*|16.6|16.4|**44.1**|30.8 ↑0.8|22.6 ↓0.1|15.8 ↑2.0|
> > > ||+MAmmoTH-2|37.4|25.7 ↑0.3|20.6 ↑0.5|26.0|22.0|22.1*|16.4|16.1|43.8|30.0 ↑0.0|22.5 ↓0.2|14.1 ↑0.3|
> > > ||+Magpie-v0.3|36.8|25.9 ↑0.5|20.7 ↑0.6|26.8|22.2|22.6|16.2|15.5|**44.1**|30.8 ↑0.8|22.7 ↑0.0|**16.4 ↑2.6**|
> > > ||+DeepSeek-R1-Distill|38.1|27.0 ↑1.6|22.1 ↑1.1|28.4|22.7|22.5|17.3|15.1|43.7|33.2 ↑3.2|24.3 ↑1.6*|15.1 ↑1.3|
> > > ||+Dart-prop|38.0|**28.9 ↑3.5***|**23.7 ↑3.6***|30.7*|24.8*|**25.5***|**19.5***|**17.4***|43.6|**33.6 ↑3.6***|**24.5 ↑1.8***|14.8 ↑1.0|

---

### Official Review · Reviewer_wWNd · 2025-03-25

**Overall Recommendation:** 3

**Summary:**

This paper investigates the impact of integrating math-specific LLMs into VLMs through model merging. The experimental results demonstrate that this approach effectively transfers reasoning abilities from math-specific LLMs to VLMs in a training-free manner. Furthermore, the authors conduct extensive experiments to analyze the distribution of perception and reasoning abilities within the model’s parameter space by modifying different layers. Their findings indicate that perception capabilities primarily originate from the early layers, while reasoning abilities are predominantly derived from the middle-to-late layers. Additionally, they conclude that merging with reasoning models enhances the mathematical reasoning ability across most layers of the model.

**Claims And Evidence:**

Most claims are supported by corresponding empirical evidence. However, the authors assert that "after merging, we observe that all layers begin to contribute to reasoning, whereas the distribution of perception abilities across layers remains largely unchanged." I could not find any text discussing this observation in further detail.

**Essential References Not Discussed:**

As far as I know, most essential references have been appropriately discussed in the manuscript.

**Experimental Designs Or Analyses:**

The experimental results (e.g., Tables 2 and 3) and analysis (e.g., Figures 4 and 5) are detailed and well-presented.  I appreciate the thorough analysis of Figures 4 and 5.  However, I find the conclusion stated in line 375—"Merging with Reasoning Models Enhances Almost All Layers’ Math Reasoning Ability"—to be inconsistent with Finding 4, which states, "The contribution of the middle-to-late MLP layers and almost all attention layers to math reasoning has increased."  The observation that more layers show an impact on accuracy does not necessarily imply an improvement in their reasoning ability.  Instead, I believe this section presents a more robust analysis rather than definitive evidence of enhanced reasoning ability.

**Methods And Evaluation Criteria:**

Model merging is a common and effective approach in the field of vision-language models.

**Other Comments Or Suggestions:**

Some of the character sizes of figures, particularly in Figures 4 and 5, appear to be too small, which may make them difficult for readers to interpret.

**Other Strengths And Weaknesses:**

Strengths:

The paper provides a comprehensive analysis of integrating the reasoning abilities of large language models (LLMs) into vision-language models (VLMs). It includes detailed performance comparisons and insightful visualization analyses, effectively highlighting the impact of this integration.

Weaknesses:

A major concern is the experimental setting. Although merging LLMs with extensive mathematical knowledge into VLMs is rational, given that math abilities are crucial for logical reasoning, it is unclear why the authors did not explore the use of other powerful logical LLMs.

**Questions For Authors:**

I am curious about the performance improvements observed in Figures 4 and 5 when modifying or replacing the model parameters. Intuitively, such changes would be expected to either maintain or decrease performance rather than enhance it. Could you clarify this discrepancy?

**Relation To Broader Scientific Literature:**

This paper demonstrates that integrating math-specific LLMs can consistently enhance the reasoning capabilities of VLMs, offering valuable insights into the field of multimodal integration.

**Theoretical Claims:**

This paper is empirically focused, with minimal theoretical equations or claims.

---

> ### Author Rebuttal · Authors · 2025-04-01
>
> We appreciate your thorough review and detailed comments! We address your questions below.
>
> Q1:
> >The authors assert that "after merging, we observe that all layers..whereas the distribution of perception abilities across layers remains largely unchanged." I could not find any text discussing this observation.
>
> Thanks for your advice! This observation is briefly discussed in Section 4 (line 376–383) in paper. After merging with Dart, nearly all layers exhibit a larger performance drop in math reasoning tasks compared to LLaVA, suggesting that reasoning capabilities are more widely integrated across layers post-merging.
>
> The statement about “distribution of perception abilities across layers remains largely unchanged” is an **implicit deduction**.
> - the higher influence of all layers in math reasoning task (Math subset of MathVista) implies that reasoning is layered on top of perception abilities, rather than replacing them. Since this task requires both perception and reasoning, the observation that early (perception-focused) layers continue to show sensitivity by masking supports this interpretation. If reasoning overrides perception, the influence of early layers would diminish—but it is not observed.
> - Besides, in general VQA tasks (see left side of  ⑤ and ⑦ vs. ⑥ and ⑧), we observe no substantial shift in the behavior of individual layers after merging. This further supports the idea that perception-related representations remain stable.
>
> We appreciate your feedback and will revise the discussion in Section 4 of the next version to clarify this point.
>
> Q2:
> >More layers show an impact on accuracy does not necessarily imply an improvement in their reasoning ability.
>
> Thank you for your thoughtful feedback! We agree that the original phrasing may convey a stronger claim than intended. We will revise it to clarify that our analysis suggests a potential correlation, not conclusive evidence. Specifically, we hypothesize that the presence of more layers may correlate with improved reasoning performance, as the amount of reasoning-related latent knowledge stored across layers may also increase.
>
> Q3:
> > it is unclear why the authors did not explore the use of other powerful logical LLMs.
>
> Thank you for your valuable feedback. We share your interest in exploring the generalization of reasoning capabilities across domains, including logical reasoning.
>
> - Our paper already includes diverse LLMs trained on various reasoning tasks. Specifically, we incorporated Magpie, which covers planning and coding tasks (see [1] for details), and Mammoth2, a model trained on various reasoning domains including Socratic reasoning - a form of logical reasoning. The summary of training data as shown below.
>
> |Domain|Size|Subjects|
> |-|-|-|
> |MathStackExchange|1484630|Mathematics|
> |ScienceStackExchange|317209|Physics, Biology, Chemistry, Computer Science|
> |Socratic|533384| Mathematics, Science, Humanities (Logical)|
>
> Performance results for both are at table 1 in our paper. They show only minor gains, suggesting that combining general-purpose reasoning models remains insufficient for strong performance on math-heavy benchmarks.
> - To explore further on logical reasoning, we add the experiment by fine-tuning a LLaMA3-8B on LogiCoT [3] (a logical chain-of-thought dataset) and merging with LLaVA. As results shown below, this pure logic-focused approach helps the math-reasoning tasks, indicating good generalization ability, we will update these experiment results in our paper.
> |**Model**|**Method**|**MathVista**|||**MathVerseBenchmarks**||||||**MV**|
> |-|-|:-:|:-:|:-:|:-:|:-:|:-:|:-:|:-:|:-:|:-:|
> |||All|General|Math|Overall|T-D|T-L|V-I|V-D|V-O|||
> |LLaVA|Baseline|37.4|51.7|25.4|20.1|25.9|20.8|21.1|16.5|16.0|13.8|
> ||+logic|37.0|49.1|26.7↑1.3|23.5↑3.4|30.5|25.0|26.4|20.3|15.1|11.2↓2.6|
>
> Q4:
> >Some of the character sizes of figures appear to be too small.
>
> Thank you for the feedback! We will enlarge the font sizes in these subfigures to ensure clarity  in the revised version.
>
> Q5:
> >I am curious about the performance improvements observed in Figures 4 and 5 when modifying or replacing the model parameters.
>
> For instance, in the math-targeted VQA tasks ① and ③ (right panel) shown in Figure 4, performance notably improves after masking by LLaMA, indicating that the image SFT training—designed to equip LLaVA with image embedding comprehension—**may overemphasize learning image perception**, and cause forgetting issues to diminish LLaMA’s inherent mathematical reasoning skills. Restoring the original parameters recovers this capability, highlighting the need to preserve strong reasoning skills in VLMs to counteract domain shift introduced by image training.
>
> [1] Magpie: Alignment Data Synthesis From Scratch By Prompting Aligned Llms With Nothing, Zhangchen et al. ICLR 2025.
> [2] MAmmoTH2: Scaling Instructions from the Web. Xiang et al. NeurIPS 2024.
> [3] LogiCoT: Logical Chain-of-Thought Instruction Tuning. Liu et al.EMNLP 2023

---

> > ### Comment · Reviewer_wWNd · 2025-04-05
> >
> > Thank you for the authors' detailed response. I am still curious, however, about which type of LLMs is more effective for improving multi-modal reasoning and perceptual abilities—using logic-focused LLMs or math-based LLMs?

---

> > > ### Author Response · Authors · 2025-04-06
> > >
> > > Thank you for your thoughtful response! To enhance reasoning capabilities for VLMs, we recommend merging with math-based LLMs for the following reasons:
> > >
> > > - Merging with pure-logical LLM is not as effective as pure-math LLM :
> > >
> > >   To make our conclusions clearer, we put together our additional experiment results for LogiCoT [1] with the results for Dart-Prop in the table below.
> > >
> > >   | **Model** | **Method** | **MathVista** |     |     | **MathVerseBenchmarks** |     |     |     |     |     | **MV** |
> > >   |-----------|------------|:-------------:|:---:|:---:|:------------------------:|:---:|:---:|:---:|:---:|:---:|:------:|
> > >   |           |            | All           | General | Math | Overall | T-D  | T-L  | V-I  | V-D  | V-O  |      |
> > >   | LLaVA     | Baseline   | 37.4          | 51.7    | 25.4 | 20.1    | 25.9 | 20.8 | 21.1 | 16.5 | 16.0 | 13.8 |
> > >   |      | +logic     | 37.0          | 49.1    | 26.7 ↑1.3 | 23.5 ↑3.4 | 30.5 | 25.0 | 26.4 | 20.3 | 15.1 | 11.2 ↓2.6 |
> > >   ||+Dart-prop|38.0|48.7|28.9 ↑3.5|23.7 ↑3.6|30.7|24.8|25.5|19.8|17.4|14.8 ↑1.0|**
> > >
> > >   We observe that while merging with a purely logical LLM offers benefits on certain benchmarks, such as MathVerse and the Math subset of MathVista, it still lags behind merging with the math-specific LLM, Dart-Prop in almost all cases.
> > >
> > > - Merging with general-purpose reasoning LLMs—which contain both logical and mathematical domain knowledge—also shows to be less effective.
> > >
> > >   As shown in Table 2, our experimental results indicate that general-purpose reasoning LLMs such as Magpie and Mammoth2 also show limited improvement compared with Dart. This indicates that merging with an LLM trained on mixed knowledge—including logical, mathematical, and other reasoning domains—does not outperform merging with an LLM specialized in mathematical reasoning.
> > >
> > > Regarding perception ability, we believe that merging can preserve core perceptual capabilities but does not lead to any improvement of perception. This is because the injected knowledge from LLMs primarily targets reasoning and does not involve perception-related information.
> > >
> > > Thank you again for your time and valuable comments. We will incorporate these discussions into our paper according to your suggestions.
> > >
> > > [1] LogiCoT: Logical Chain-of-Thought Instruction Tuning. Liu et al.EMNLP 2023

---

### Decision · Program_Chairs · 2025-05-01

**Decision:**

Accept (poster)

**Comment:**

The submission investigates merging vision-language models (VLMs) with large language models (LLMs) specialized in reasoning, demonstrating effective transfer of reasoning capabilities into VLMs via a simple linear merging method. Reviewers acknowledged the paper’s comprehensive experimental validation and insightful layer-wise analyses revealing that perception capabilities predominantly reside in early model layers, whereas reasoning emerges from middle-to-late layers. The authors successfully addressed most reviewer concerns through detailed rebuttals, providing additional experimental validations, such as incorporating Qwen-based models and significance testing. However, reviewers highlighted limitations, including the modest novelty of the merging technique itself and the scope limited primarily to math-based reasoning models. Despite these, the reviewers collectively agreed the paper makes valuable contributions in interpretability, clearly demonstrating the utility of cross-modal model merging for understanding internal model dynamics. Considering the detailed responses and additional results provided, the reviewers leaned towards acceptance, emphasizing the meaningful empirical insights presented.